# RNA binding protein FXR1-miR301a-3p axis contributes to p21$^{WAF1}$ degradation in oral cancer

**Mrinmoyee Majumder, Viswanathan Palanisamy**\*

Department of Biochemistry and Molecular Biology, College of Medicine, Medical University of South Carolina, Charleston, SC, United States of America

\* visu@musc.edu

**Data Availability Statement:** All small RNA-seq data files are available from the Gene Expression Omnibus database accession number GSE117031.

**Funding:** This work is supported by NIH grants DE025920 to V.P. Supported in part by pilot

## Abstract

RNA-binding proteins (RBPs) associate with the primary, precursor, and mature micro-RNAs, which in turn control post-transcriptional gene regulation. Here, by small RNAseq, we show that RBP FXR1 controls the expression of a subset of mature miRNAs, including highly expressed miR301a-3p in oral cancer cells. We also confirm that FXR1 controls the stability of miR301a-3p. Exoribonuclease PNPT1 degrades miR301a-3p in the absence of FXR1 in oral cancer cells, and the degradation is rescued in the FXR1 and PNPT1 co-knockdown cells. In vitro, we show that PNPT1 is unable to bind and degrade the miRNA once the FXR1-miRNA complex forms. Both miR301a-3p and FXR1 cooperatively target the 3'-UTR of *p21* mRNA to promote its degradation. Thus, our work illustrates the unique role of FXR1 that is critical for the stability of a subset of mature miRNAs or at least miR301a-3p to target p21 in oral cancer.

## Author summary

RNA-binding protein FXR1 is overexpressed in multiple cancers and appears to be involved in poor patient survival. FXR1 has been previously shown to be an oncogene in head and neck and lung squamous cell carcinoma. FXR1 targets tumor suppressor genes, including p21, to promote the growth and proliferation of cancer cells. However, how FXR1 recognizes tumor suppressor genes and block their expression in cancer cells has never been established. Through FXR1-mediated small RNA sequencing, we unexpectedly found that FXR1 stabilizes miR301a-3p, which in turn targets tumor suppressor *p21* and blocks its expression in oral cancer cells. miR301a-3p level goes down in the absence of FXR1, increasing p21 to suppress the growth of oral cancer cells. We provide evidence that miR301a-3p expression is rescued by downregulation of both FXR1 and exoribonuclease PNPT1 in oral cancer cells suggesting that FXR1 acts as a stabilizing factor for miR301a-3p against PNPT1. Together, our observations explain why over-expression of FXR1 and miR301a-3p in the oral cancer patient cohort lead to the downregulation of p21 signaling. These findings indicate that FXR1 and miR301a-3p together contribute towards targetting tumor suppressors in oral cancer patients.

research funding, Hollings Cancer Center's Cancer Center Support Grant P30 CA138313 at the Medical University of South Carolina. The funders had no role in study design, data collection and analysis, decision to publish, or preparation of the manuscript.

**Competing interests:** The authors have declared that no competing interests exist.

## Introduction

RNA-binding proteins (RBPs) regulate co- and post-transcriptional gene expression. RBP Fragile X Mental Retardation protein-1 (FXR1) is involved in the transport, translation, and degradation of mRNA, and often functions by binding either AU-rich elements (ARE) or G-quartet ($G_q$) regions on the RNA [1–3]. FXR1 belongs to the Fragile X-mental retardation (FXR) family of proteins, including FMRP/FMR1 and FXR2 [4]. The proteins share the same RNA binding domains (two KH domains and the RGG box), along with a nuclear localization signal (NLS) and a nuclear export signal (NES) [5]. In addition to RNA-binding activity, FXR family of proteins also associate with free ribosomes and polyribosomes [6]. FXR1 is mostly localized in the cytoplasm [4], but like many RBPs, it can shuttle between cytoplasm and nucleus [7].

MicroRNAs (miRNAs) are short (20–24 nucleotides) non-coding RNAs involved in numerous regulatory pathways in multicellular organisms [8]. Primary miRNAs are processed into pre-miRNAs and then mature miRNAs [9] by RBPs Drosha and Dicer, respectively. Mature miRNAs are loaded onto the RNA-induced silencing complex (RISC), which directs translational repression and subsequent degradation of the target mRNAs [9, 10]. RBPs also regulate miRNA maturation and stability [11, 12]. RBPs can interact with primary, premature, and mature miRNAs to influence their expression [13], often in a tissue-specific context [14, 15]. RBPs recognize and bind to specific premature or mature-miRNA sequences and secondary structures to regulate their expression and function [16]. Mature miRNA turnover is also highly regulated [17–20], although the mechanistic basis for miRNA stability remains unknown. MiRNAs can behave as oncogenes by targeting tumor suppressor mRNAs and promoting oncogenesis [21, 22]. For example, expression of the miR-17-92 cluster is found to be up-regulated in osteosarcoma and connected to the deregulation of genes involved in differentiation, cell cycle control, and apoptosis [23]. MiRNAs can also have both oncogenic and tumor-suppressor properties depending on the tissue type. For example, miR-181 acts as an oncogenic miRNA in papillary thyroid cancer [24], whereas in brain cancer it exhibits tumor suppressor activity [25].

FXR family proteins associate with miRNAs and the RNAi pathway [3, 26–30]. For example, FXR1 associates with RISC protein Argonaute-2 (AGO2) in a miRNP (ribonucleoprotein) complex and regulates mRNA translation [3]. FMRP interacts with a subset of miRNAs and regulates their expression in *Xenopus laevis* [28], and FMRP knockout mice show alterations in the expression of a subset of miRNAs [31]. The finding of an active interaction between FXR1 and AGO2 suggests that either RBPs are taken to substrates by the RNAi complex [26], or that FXR1 first aims for the G-quartet/ARE [1], and then recruit RISC proteins [29]. Though FXR1 is associated with AGO2 and miRNAs [32, 33], it is not known if these interactions are direct or if they are essential for RNAi-mediated mRNA degradation.

FXR1 is over-expressed in multiple cancers, including head and neck (HNSCC) and lung squamous cell carcinomas (LSCC) [34, 35]. Loss of FXR1 in LSCC induces apoptosis [35] and promotes cellular senescence in HNSCC [34]. Our previous study showed that FXR1 binds to the 3'-UTR of the tumor suppressor *p21* (CDKN1A, WAF1, CIP1) and targets it for decay, thereby bypassing senescence and promoting HNSCC growth and proliferation [34]. However, how FXR1 promotes *p21* degradation remains unclear. Here, we show that FXR1 protects miR301a-3p degradation from exoribonuclease activity. In the absence of FXR1, mature miR301a-3p undergoes an exonuclease Polyribonucleotide Nucleotidyltransferase 1 (PNPT1) mediated decay. We also show that miR301a-3p targets *p21*, and FXR1 knockdown leads to down-regulation of miR301a-3p and up-regulation of p21 mRNA and protein levels in multiple oral cancer cells. This phenomenon may explain why over-expression of FXR1 and miR301a-3p in an HNSCC cohort leads to the downregulation of p21 signaling. These findings

indicate that the use of FXR1 inhibitors along with miR301a-3p anti-miR oligonucleotide therapy, combined with chemotherapy, can be a better therapeutic strategy to treat HNSCC patients.

## Results

### FXR1 modulates the expression of a subset of miRNAs in oral cancer cells

Previously, we have shown that FXR1 targets *p21* for its degradation and consequently to bypass cellular senescence and promote the proliferation of oral cancer cells [34]. To investigate whether FXR1-mediated *p21* degradation requires miRNA- and RNA-induced silencing complex, we identified the miRNAs that are controlled by FXR1 in oral cancer cells. Analyses of high-throughput sequencing (HTS) of total small RNAs (<200 bases) from shControl and shFXR1-(TRCN0000159153)-treated UMSCC74B oral cancer cells revealed 254 high confidence peaks (discovery rate FDR≤0.4) (GEO accession# GSE117031), (**Fig 1**A). We selected six candidate miRNAs: miR301a-3p, miR29b-3p, miR204-5p, miR98-5p, miR125-5p, and miR30c-5p, based on their differential expressions pattern in FXR1 knockdown UMSCC74B cells. To determine whether FXR1-targeted miRNAs were differentially expressed in HNSCC, we utilized The Cancer Genome Atlas (TCGA) data for Head and Neck cancer utilizing the University of California Santa Cruz (UCSC) functional genomics browser, XENA. The dataset contains samples from 604 patients. XENA [36] showed a significant up- and down-regulation of miR301a-3p and miR-204-5p, respectively, in the HNSCC primary and metastatic tumor samples compared to normal tissues (**S1A Fig**).

To validate the RNA-seq results, we assayed the expression of the six miRNAs in FXR1-silenced oral cancer cells. Quantitative RT-PCR (qRT-PCR) showed a significant reduction in miR301a-3p and miR29b-3p in FXR1 knockdown cells compared to shControl-treated cells (**Fig 1**B). Expression of the six miRNAs was further tested in two other oral cancer cell lines, UMSCC11A, and UMSCC74A, and a lung cancer cell line, A549 (**S1B–S1D Fig**). The data indicated that miR301a-3p and miR29b-3p were significantly down-regulated in FXR1 knockdown cells compared to control cells. Based on these observations, we conclude that miR301a-3p is a possible target of FXR1 that, in turn, targets p21/CDKN1A (Targetscan). In our previous report [34], we showed that FXR1 targets p21 mRNA for degradation. Here, we wanted to see if FXR1 involves miRNA (s) to deregulate p21. Firstly, to determine if miR301a-3p is regulated by FXR1, we used multiple oral cancer cell lines and tested the expression of miR301a in FXR1-depleted cells. The qRT-PCR analyses indicated that miR301a-3p was significantly down-regulated in FXR1-depleted multiple oral cancer cells compared to control cells (**Fig 1**C). The data suggest that FXR1 controls the mature miR301a-3p level in these cancer cells. Additional miRNA miR29b-3p also showed reduced expression in FXR1 knockdown cells compared to control cells (**Figs 1**B **and S1B–S1D**). A second shRNA was also used to illustrate that the phenomenon mentioned above is specific to FXR1 (**S1E–S1G Fig**). To verify whether FXR1 knockdown was impacting pre-miR301a, we examined the levels of pre-miR301a by qRT-PCR. The pre-miR301a level did not change in most of the FXR1-depleted cells, except Cal27, indicating that only mature miR301a-3p is dependent on FXR1 in most of the oral cancer cells. Together, our previously published results and the data presented here suggest that both FXR1 [34] and miR301a-3p are amplified in HNSCC tumor samples, and FXR1 controls the expression of mature miR301a-3p in oral cancer cells.

### FXR1 binds to miR301a-3p in vivo and in vitro

FXR family member FMRP binds to mature miRNAs in human and mouse cells [37, 38]. Therefore, we set out to determine whether FXR1 directly associates with miR301a-3p and

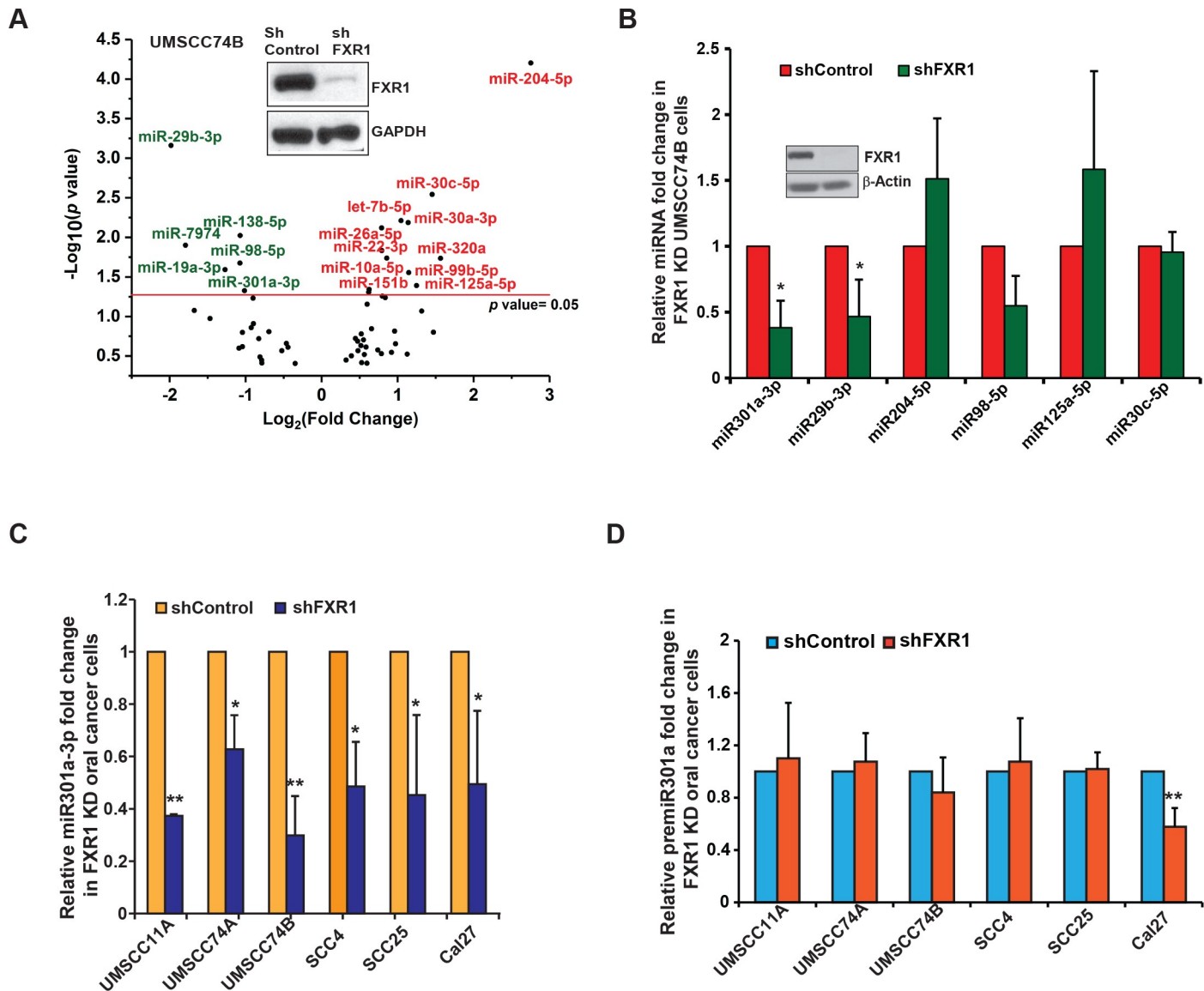

**Fig 1. FXR1 modulates the expression of a subset of miRNAs in oral cancer cells.** (**A**) Volcano plot of differential miRNA expression in UMSCC74B cells in the absence of FXR1. Red: up-regulated and green: down-regulated miRNAs in the absence of FXR1. The inset shows the knockdown efficiency of FXR1 by shRNA (TRCN0000159153) compared to a scrambled shRNA where GAPDH serves as a loading control. X-axis signifies the $\log_2$ values of the miRNA fold change where y-axis plots the $-\log_{10}$-$p$-values of the differentially expressed miRNAs in the FXR1 knockdown cells compared to control. (**B**) qRT-PCR of the altered miRNAs in FXR1 knockdown UMSCC74B cells. RNU6 serves as an endogenous control. Inset shows the knockdown efficiency of FXR1 by shRNA (TRCN0000159153) compared to a scrambled shRNA where β-Actin serves as a loading control. (**C**) qRT-PCR is showing that miR301a-3p is significantly down-regulated in all the FXR1 knockdown HNSCC cell lines. RNU6 serves as an endogenous control. (**D**) qRT-PCR is showing that premiR301a is not altered in most of the FXR1 knockdown HNSCC cell lines except Cal27. 18S rRNA serves as an endogenous control. Data from B-D represent the mean of n = 3 experiments. Statistical significance ($p$-value): $^{*}$<0.05; $^{**}$<0.005. See also S1 Fig.

stabilizes its levels in oral cancer cells. To investigate the association between FXR1 and mature miR301a-3p in oral cancer cells, we used RNA-IP (RNA immunoprecipitation) as described by us [34]. The UMSCC74B cells were subjected to RNA-IP using an FXR1 antibody, revealing significant binding of miR301a-3p to FXR1 compared to the IgG control beads (**Fig 2A**) whereas miR204-5p did not show any significant binding to FXR1 protein by the same method (**S2A Fig**). To investigate whether FXR1 directly binds to mature miR301a-3p and forms a

miRNP (miRNA ribonucleoprotein) complex, we cloned FXR1 in pET28a vector (His-tag) and purified the recombinant protein FXR1 (rFXR1). The rFXR1 was purified from a Ni-NTA affinity column, and different fractions were collected for dialysis. The purified rFXR1 protein yielded a single band in coomassie stained SDS polyacrylamide gel (**Fig 2**B). Two different antibodies further confirmed the rFXR1 protein in the pooled fractions before and after dialysis (removal of His-tag by thrombin digestion), such as His-tag and FXR1 (**S2B and S2C Fig**). To determine rFXR1 binding with miR301a-3p, we labeled the 23 bases miR301a-3p with [$\gamma^{32}$-P] ATP and incubated with rFXR1 for electrophoretic mobility shift assay (EMSA). An increased concentration of rFXR1 were incubated with radio-labeled miR301a-3p to form miRNP complex at 37˚C for 30 min (**Fig 2**C). The data demonstrates that the rFXR1 can bind to the mature miR301a-3p, consistent with the RNP-IP in vivo results. Our previous publication [34] showed that FXR1 regulates both p21 and TERC RNA to bypass senescence. We wanted to see if the downregulation of miR301a-3p independently or along with TERC RNA can have any effect on cellular senescence in UMSCC74B cells. We found that controlling both miR301a-3p and TERC RNA has affected cellular senescence in UMSCC74B cells (**S2D Fig**).

## Stability of miR301a-3p is FXR1-dependent

To determine whether miR301a-3p levels were reduced in the FXR1 knockdown cells due to increased degradation, we measured the level of miR301a-3p in control and FXR1 knockdown oral cancer cells. We measured the miR301a-3p levels in the two different oral cancer cells, which are collected every 12 hours for 72 hours post-shFXR1 transduction. We observed that shFXR1 transduction led to a reduction in FXR1 mRNA significantly at 12 hours and showed a 90% reduction at 72 hours (**Figs 3**A **and S**3A). The FXR1 target *p21* also started to increase significantly at 36 hours and showed four-fold up-regulation after 72 hours (**Figs 3**A **and S**3A). Similar trends were observed for FXR1 and p21 proteins at different time points (**Figs 3**B **and S**3B). AGO2 mRNA and protein did not change in FXR1 knockdown cells, and only a small reduction in AGO2 protein was observed in UMSCC11A cells treated with shFXR1 (**Fig 3**A, **3**B, **S**3A **and S**3B). We found that FXR1 knockdown reduced the levels of miR301a-3p significantly at 48 hours and showed a 60% reduction at 72 hours (**Figs 3**C **and S**3C) compared to control cells, indicating that miR301a-3p stability is dependent on FXR1. Consistent with miRNAs having half-lives ranging from >8 h to several days [39], we observed miR301a-3p was hardly degraded in control cells. We wanted to determine if changes in transcription levels could interfere with the miR301a-3p level in the presence and absence of FXR1. The transcription was shut down by actinomycin D treatment for 8 hrs in UMSCC74B cells stably expressing IPTG inducible shControl and shFXR1 [34], and the impact of FXR1 on miR-301a stability was evaluated (**Fig 3**D). The miR301a-3p level was reduced in actinomycin D treated cells after 8 hrs compared to DMSO treated cells; however, the miRNA level was degraded entirely in both IPTG and actinomycin D treated cells. The reduction in miR301a-3p stability upon FXR1 knockdown argues that FXR1 stabilizes miR301a-3p in cancer cells.

Argonaute proteins, especially AGO2, are the key players in miRNA processing, stability, and function [40]. To investigate whether silencing of AGO2 in these oral cancer cells altered the expression of miR301a-3p, we collected the cells at 72 hours after AGO2 knockdown, and tested the levels of miR301a-3p. We observed that FXR1 and p21 mRNA/protein did not show any appreciable changes in their expression after AGO2 knockdown (**Fig 3S, 3**D–**3F**). The level of miR301a-3p showed no regulation after 72 hours of siAGO2 post-transfected UMSCC74B cells (**S3F Fig**). This finding indicates that unlike FXR1 knockdown, mature miR301a-3p stability did not depend on AGO2, at least in the oral cancer cell lines used here. Together, these findings suggest that FXR1 controls the balance of miR301a-3p in oral cancer cells.

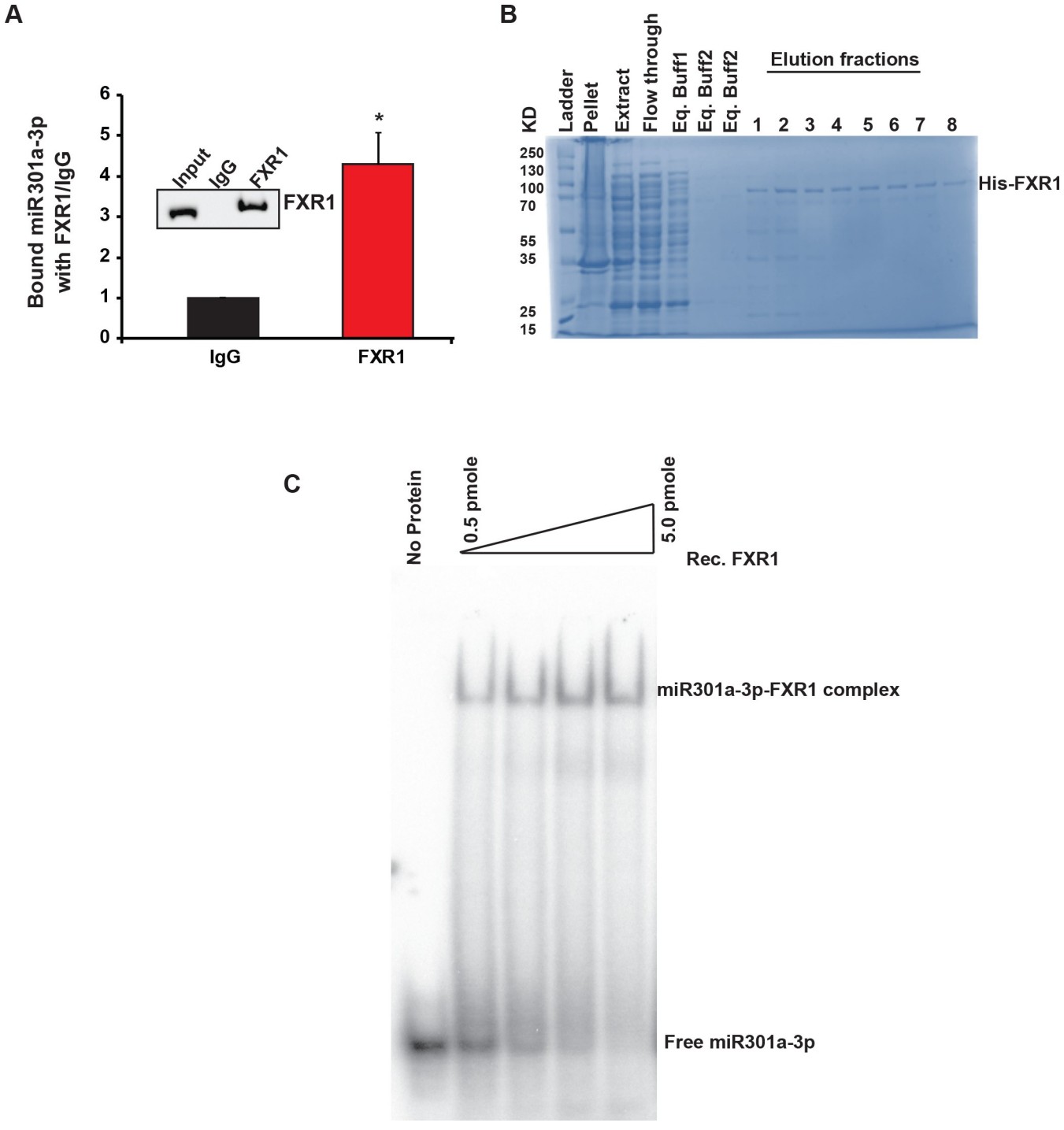

**Fig 2. FXR1 binds to miR301a-3p in vivo and in vitro.** (**A**) RNA-immunoprecipitation shows miRNA301a-3p binds to FXR1 in UMSCC74B cells compared to control mouse IgG. Both *ACTIN* and *RPS18* served as endogenous controls. FXR1 antibody pull-down efficiency by immunoprecipitation is shown in the inset. (**B**) SDS-PAGE showing rec. FXR1 purification from the Ni-NTA column. (**C**) EMSA with 5'-labeled miR301a-3p and rec. FXR1 protein. 0.5 pmole of [γ-$^{32}$P] ATP labeled miRNA was mock-treated or mixed with increasing concentration of recombinant FXR1 protein and incubated at 37˚C for 30 min. Free RNA and miRNP complexes are shown in the figure.

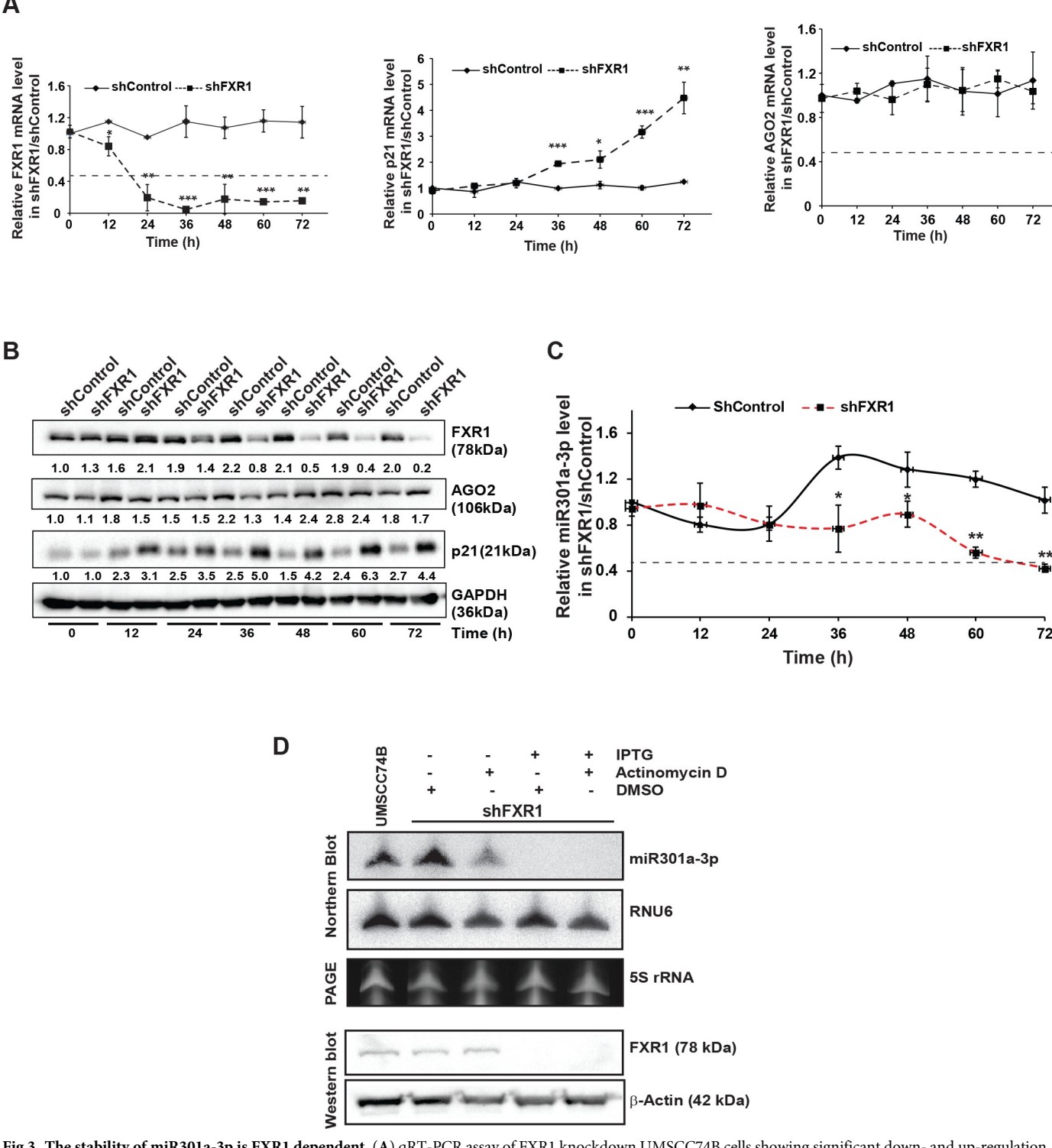

**Fig 3. The stability of miR301a-3p is FXR1 dependent.** (**A**) qRT-PCR assay of FXR1 knockdown UMSCC74B cells showing significant down- and up-regulation of *FXR1* and *p21*, respectively, compared to control. *AGO2* did not show any change after the FXR1 knockdown. Both *ACTIN* and *RPS18* served as endogenous controls. (**B**) Western blot analyses of FXR1, p21, and AGO2 from UMSCC74B cells collected at different time points as mentioned after shFXR1 transduction. GAPDH serves as a loading control. (**C**) qRT-PCR assay of FXR1 knockdown UMSCC74B cells showing significant miR301a-3p decay from 48hrs compared to control. Cells were collected at the designated time points after shRNA transduction. RNU6 serves as an endogenous control. (**D**) UMSCC74B cells stably expressing IPTG inducible shControl and shFXR1 were treated with 1mM IPTG for 72 hrs, followed by actinomycin D (ActD) or DMSO treatment for 8 hrs. Total RNA was prepared from all samples for northern hybridization. RNU6 serves as an endogenous control whereas total sample loading was shown by the 5s rRNA

level from the PAGE. The same samples were used for western blot to show the knockdown of FXR1. Statistical significance (*p*-value): *<0.05; **<0.005; ***<0.0005. See also S3 Fig.

## PNPT1 degrades miR301a-3p in the absence of FXR1

RBPs associate with nascent RNAs and protect them from exonuclease mediated degradation [41]. To determine whether FXR1-mediated miR301a-3p stabilization is coordinated by exonucleases that are present or aberrantly expressed in HNSCC, we tested their levels in oral cancer cells and A549 cells in comparison with normal human oral keratinocyte (HOK). We tested the mRNA levels of the three significant exoribonucleases that function in miRNA decay: 5'-3' Exoribonuclease 1 (XRN1), 5'-3' Exoribonuclease 2 (XRN2), and 3'-5' RNA exonuclease polyribonucleotide nucleotidyltransferase 1 (PNPT1) [42, 43]. XRN1 and XRN2 dissociate many miRNAs from miRISC and degrade them in the 5'-to-3' direction, whereas PNPT1 degrades a small subset of miRNAs in the 3'-to-5' direction [43]. As shown in **Fig 4**A, *XRN1* was expressed at significantly lower levels in the oral cancer cell lines compared to normal HOK cells. *XRN2* also exhibited low expression in most cell lines compared to HOK. Unlike *XRN1/2*, *PNPT1* was significantly up-regulated in all the cell lines tested compared to HOK cells along with FXR1 (**Fig 4**A). To further confirm the cell line results, we used UCSC XENA and found up-regulated *FXR1* (p = 0.0139) and *PNPT1* (p = 0.00017) in primary tumors, compared to normal solid tissue (604 total HNSCC patients' samples) by one-way ANOVA (**S4A Fig**). Together, these findings indicate that PNPT1 is overexpressed in HNSCC. Previous studies on PNPT1-mediated miRNA degradation suggests that PNPT1 degrades mature miR221 and is involved in the growth and proliferation of melanoma cells [44]. We therefore asked whether PNPT1 co-knockdown with FXR1 can rescue the reduction in miR301a-3p levels that is observed in FXR1 KD cells alone. We found that PNPT1 co-knockdown with FXR1 was sufficient to block the reduction of miR301a-3p levels (**Figs 4B and S4C**). Additionally, the miR301a-3p levels showed a slight but significant increase (1.175 ± 0.091) after the double co-knockdown of FXR1 and PNPT1 in UMSCC11A cells (**S4C Fig**). These observations suggest that PNPT1-mediated degradation of miR301a was blocked by FXR1 and, in the absence of FXR1, PNPT1 is recruited to degrade miR301a-3p.

To investigate if miR301a-3p directly impacts the expression of *p21* through the FXR1-PNPT1 axis, we tested whether the FXR1 and PNPT1 double knockdown rescue p21 mRNA and protein. As expected, FXR1 knockdown increased the levels of p21 mRNA and protein (**Fig 4C, 4D**, **S4D and S4E**), where the miR301a-3p level was compromised in these cells (**Figs 4B** and **S4C**). However, we observed no change in the levels of p21 mRNA with PNPT1 knockdown alone (**Figs 4C, 4D**, **S4D and S4E**). Moreover, a double knockdown of FXR1 and PNPT1 reduced the expression of p21 mRNA and protein compared to the FXR1 KD cells (**Figs 4C, 4D and S4E**).

The previous results suggest that FXR1 may protect miR301a-3p from PNPT1 exoribonuclease mediated degradation. To check if PNPT1 is unable to target miR301a-3p when it is bound to FXR1 protein, we conducted in vitro miR301a-3p-FXR1 and miR301a-3p-PNPT1 binding assay using recombinant FXR1 and PNPT1. Like RIP assay in vivo (**Fig 2A**), recombinant FXR1 forms complex with miR301a-3p in vitro (**Figs 2C** and **4E**, lane 2). However, incubation of miR301a-3p with recombinant PNPT1 degraded the miRNA entirely (**Fig 4E**, lane 3, and 6). In contrast, upon sequential binding with FXR1 and miR301a to form an mRNP complex, followed by PNPT1 addition, miR301a-3p was protected in the miRNP complex (**Fig 4E**, lane 4 and 7). To show that FXR1 recombinant protein binds specifically to miR301a-3p by EMSA, we have also use miR204-5p as shown before for in vivo binding (**S2A Fig**). miR204-5p

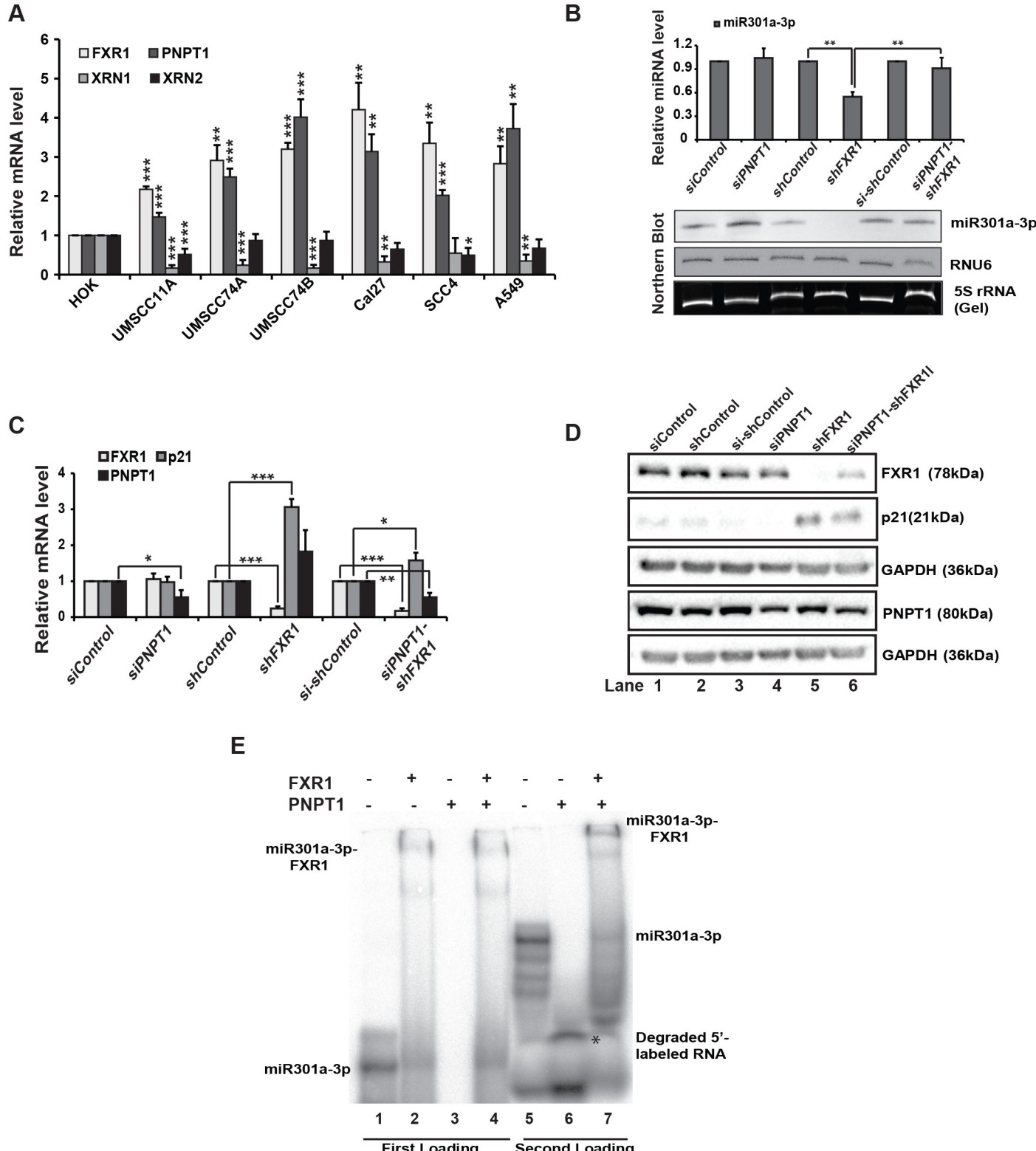

**Fig 4. PNPT1 is overexpressed in HNSCC that degrades miR301a-3p in the absence of FXR1.** (**A**) qRT-PCR analysis of *FXR1*, *XRN1*, *XRN2*, and *PNPT1* in HNSCC cell lines compared to primary line HOK along with lung cancer cell line A549. Both *ACTIN* and *RPS18* served as endogenous controls. (**B**) (up) qRT-PCR analyses to test the expression of miR301a-3p in UMSCC74B cells under individual and double knockdown of FXR1 and PNPT1. RNU6 served as an endogenous control. (down) Northern hybridization of total RNA isolated from UMSCC74B cells under individual and double knockdown of FXR1 and PNPT1 shows the levels of miR301a-3p. RNU6 is used as an endogenous control. 5S rRNA is used as a loading control. (**C**) qRT-PCR analysis of *FXR1*, *p21*, and *PNPT1* in UMSCC74B cells under individual and double knockdown of FXR1 and PNPT1. Both *ACTIN* and *RPS18* served as endogenous controls. (**D**) Western blot analysis of FXR1, p21, and

PNPT1 in UMSCC74B under individual and double knockdown of FXR1 and PNPT1. GAPDH serves as an endogenous control. (**E**) EMSA with 5'-labeled miR301a-3p with rec. FXR1 and rec. PNPT1 proteins. L1: RNA only, L2: RNA + FXR1, L3: RNA + PNPT1, L4: RNA + FXR1 (5 pmole) for 15 min, followed by PNPT1 (5 pmole) for another 15 min. L5-7: samples are loaded 2 hours after the first loading of from L1,3, and 4 to visualize the bottom of the gel. Statistical significance (*p*-value): *<0.05; **<0.005; ***<0.0005. See also S4 Fig.

did not bind to rFXR1 protein in vitro (**S4F Fig**) and also did not get degraded by recombinant PNPT1 protein (**S4F Fig**). The data here provides evidence that FXR1 bound with miR301a-3p is protected from PNPT1 mediated degradation in oral cancer cells.

### *p21* is a target of miR301a-3p

To determine the biological role of FXR1-mediated regulation of miR301a, we examined whether miR301a-3p targets mRNA(s) that are known FXR1 targets in oral cancer cells. Of the down-regulated miRNAs in FXR1-depleted cells, miR301a-3p was predicted (TargetScan) to bind to the 3'-UTR of *p21* (**Fig 5**A). Our previous findings suggest that FXR1 knockdown cells exhibit DNA damage, G0/G1 cell cycle arrest, increased p21 mRNA and protein levels, and cellular senescence [34]. Hence, we expected that the manipulation of the miR301a-3p might impact p21 levels in oral cancer cells. The following observations argue that gain- and loss-of-function of miR301a-3p alters p21 levels by relieving miRNA-mediated translational control of *p21*. First, to investigate whether miR301a-3p targets and alters *p21* levels in vivo, we transfected HNSCC cells UMSCC74B (**Fig 5**B), UMSCC74A (**S5A Fig**), and LSCC A549 (**S5G Fig**) with anti-miR301a-3p. The transfected cells show a significant reduction in miR301a-3p expression and, subsequently, increased expression of p21 protein compared to cells transfected with the antimiR scrambled control (**Figs 5**C, **S5**B **and S5**H). Similarly, others have shown that inhibition of miR301a-3p increases the levels of p21 and severely blocked prostate cancer cell growth, both in vitro and in vivo [45]. Second, our previous findings demonstrate that FXR1 knockdown leads to an increased p21 3'-UTR luciferase activity [34]. Hence, we expressed a Renilla luciferase plasmid carrying the full-length *p21* 3'-UTR and tested its expression in anti-miR301a-3p treated cells. The luciferase expression was significantly increased in the cancer cell lines treated with the antimiR301a-3p compared to controls (**Figs 5**D, **S5**C **and S5**I). Third, transfection of a miR301a-3p mimic into the cancer cell lines (**Figs 5**E, **S5**D **and S5**J) resulted in a significant down-regulation of p21 protein (**Fig 5**F, **S5**E **and S5**K) compared to the mimic scrambled controls. Finally, to investigate if there is a direct interaction between miR301a-3p and *p21* 3'-UTR, we made mutations (mut1 and mut2) in the two predicted *p21* 3'-UTR miR301a-3p binding sites (**Fig 5**A). The oral cancer cell lines were transfected with the luciferase *p21* full-length WT 3'-UTR or either of the two mutants (mut1 and mut2), together with the miR301a-3p mimic, or a corresponding control. Cells expressing WT *p21* 3'-UTR showed significantly reduced expression in the presence of the miR301a-3p mimic compared to the scrambled miRNA mimic but had little effect on the luciferase activity of mutant *p21* 3'-UTR mut1 and mut2 in both oral cancer cell lines (**Figs 5**G **and S5**F). Altogether, these results indicate that miR301a-3p targets *p21* and is responsible for its translational repression in oral cancer and A549 cells. Our data provide a molecular explanation for why oral cancer patients have loss-of-function in p21 and poor patient survival [46], where FXR1 and miR301a-3p are overexpressed in HNSCC patients.

### FXR1 and miR301a-3p cooperatively target and repress p21 expression

Similar to miR301a-3p, several miRNAs involved in oncogenesis, including miR-106b, miR-103, miR-107, miR-193, miR-29b, and miR-320, are down-regulated by PNPT1 [44]. We also observed that miR-29b was down-regulated along with miR301a-3p in FXR1 knockdown cells

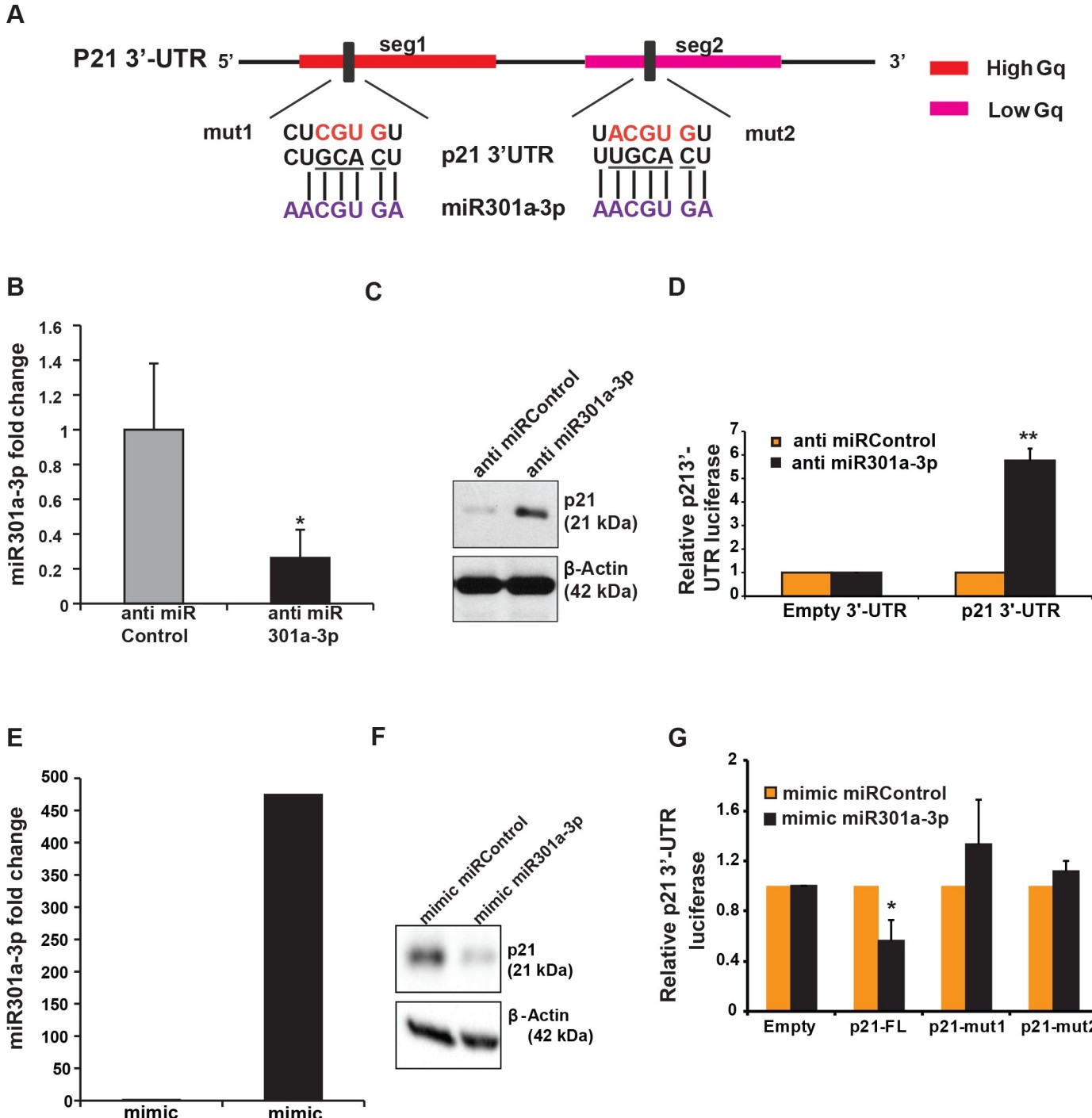

**Fig 5. miR301a-3p targets *p21* mRNA to repress translation.** (**A**) A schematic showing miR301a-3p binding sites on *p21* 3'UTR. Gq signifies the G-quadruplex regions on *p21* 3'-UTR for FXR1 binding as well. Gq regions on human p21 3'-UTR is segregated into high Gq (red) and low Gq (pink) bearing regions. miR301a-3p has a binding site in each of the high and low Gq region. Binding sites are shown in the figure with miRNA base-pairing. Mutant bases (to disrupt the miRNA-mRNA duplex) are shown in red. (**B**) qRT-PCR analyses of miR301a-3p in UMSCC74B cells treated with miRNA inhibitor and a scrambled control. RNU6 serves as an endogenous control. (**C**) p21 protein is up-regulated in miR301a-3p inhibitor transfected UMSCC74B cells. β-Actin serves as a loading control. (**D**) p21 3'UTR luciferase activity is significantly up-regulated in the presence of miR301a-3p inhibitor in UMSCC74B cells compared to the scrambled control transfected cells. Cells were collected forty-eight hours post- transfection with miRNA control and 301a-3p inhibitor along with empty 3'UTR luciferase plasmid and wild type *p21* 3'UTR, the lysates were analyzed for luciferase activity using a luminometer. The empty 3'UTR luciferase plasmid served as a transfection and loading control. Values here are the means ± SD from three independent experiments by using an unpaired two-sample t-test. (**E**) Expression of miR301a-3p in UMSCC74B cells

treated with miRNA mimics compared to scrambled control. RNU6 served as an endogenous control. (**F**) p21 protein is down-regulated in miR301a-3p mimic treated cells compared to control. β-Actin serves as a loading control. (**G**) p21 3'UTR luciferase activity is significantly down-regulated in the miR301a-3p mimic treated UMSCC74B cells compared to the scrambled mimic treated cells. However, the luciferase activity in mut1 and mut2 is seg1 and the seg2 region of miR301a-3p binding does not change after mimic transfection. Experiments were performed as described in (D). Statistical significance (*p*-value): *<0.05; **<0.005; ***<0.0005. See also S5 Fig.

(**Fig 1**A), suggesting that FXR1 is vital for a subset of oncogenic miRNA levels in oral cancer cells to regulate specific mRNA targets. To investigate whether miR301a-3p alone or in combination with FXR1 is needed to attenuate *p21*, we transfected miR301a-3p mimic in FXR1 knockdown UMSCC74B cells. First, we observed down-regulation of miR301a-3p in FXR1 knockdown cells compared to both controls, along with these cells expressing scrambled miRNA mimic (**Fig 6**A). However, the miR301a-3p levels showed a significant reduction when ectopically expressed from a mimic in FXR1 knockdown cells compared to control cells (**Fig 6**A). This observation suggests that FXR1 is required to maintain a steady level of miR301a-3p at all circumstances, which includes overexpression of miR301a-3p in trans.

Next, to determine whether overexpression of miR301a-3p under the gain- or loss-of-function of FXR1 affects its target p21 levels, we tested the expression of p21 mRNA and protein. We found that p21 levels were significantly up-regulated in FXR1 knockdown cells by both qRT-PCR and western blot (**Fig 6**B and **6**C, lanes 1 and 2). However, ectopic expression of miR301a-3p did not substantially reduce p21 mRNA or protein level in FXR1 knockdown cells (**Fig 6**B and **6**C, lane 6), compared to control cells (**Fig 6**B and **6**C, lane 5). This observation shows that FXR1 is critical for miR301a-3p mediated repression of p21 in oral cancer cells.

To delineate if an increase in substrate (*p21*) concentration in the FXR1 knockdown cells can consume or reduce the pool of miR301a-3p, we ectopically expressed p21 in oral cancer cells and tested the expression of miR301a-3p. Overexpression of p21 (**Fig 6**D and **6**E) did not alter the levels of FXR1 (**Fig 6**D and **6**E) or miR301a-3p (**Fig 6**F) in the oral cancer cells tested here, suggesting that consumption of miR301a-3p or feedback regulation of p21-mediated control of either miR301a or FXR1 was not occurring in the oral cancer cells. Finally, to validate our observations in cell lines in vivo, we have used a mouse xenograft model where we injected the mice with UMSCC74B cells stably expressing IPTG inducible shControl and shFXR1. 50% of each of male and female mice (total 12) were divided and injected with the UMSCC74B cells stably expressing IPTG inducible shControl (left) and shFXR1 (right) on each of the flanks. 50% of the population was treated with 10mM IPTG/5% sucrose in the drinking water, starting the following day of injection. And the rest were treated after the first tumor nodule was observed. As can be seen from the results in (**Fig 6**G) the first group (both male and female) failed to grow any tumor in FXR1 knockdown side compared to the control side, whereas in the second batch, the male mice had the nodule on the shFXR1 injected flank which did not grow up to the control flank. We did not observe any bulge in the second batch of female mice after dissection. The tissue protein (FXR1 and p21) data correlate with our cell line data, and so is miR301a-3p levels from the three small tumors (M#4–6) compared to control. These observations argue that FXR1 knockdown increases p21 levels by augmenting miR301a-3p-mediated translation repression. Based on the above data, a model representing FXR1 mediated stabilization of miR301a-3p from exonuclease PNPT1 to regulate p21 translation in cancer cells (**Fig 7**).

## Discussion

### FXR1 regulates specific miRNA stability in human oral cancer cells

In this report, we describe a pathway of miRNA regulation in which FXR1 represses PNPT1-mediated degradation in oral cancer cells. Numerous observations in the manuscript

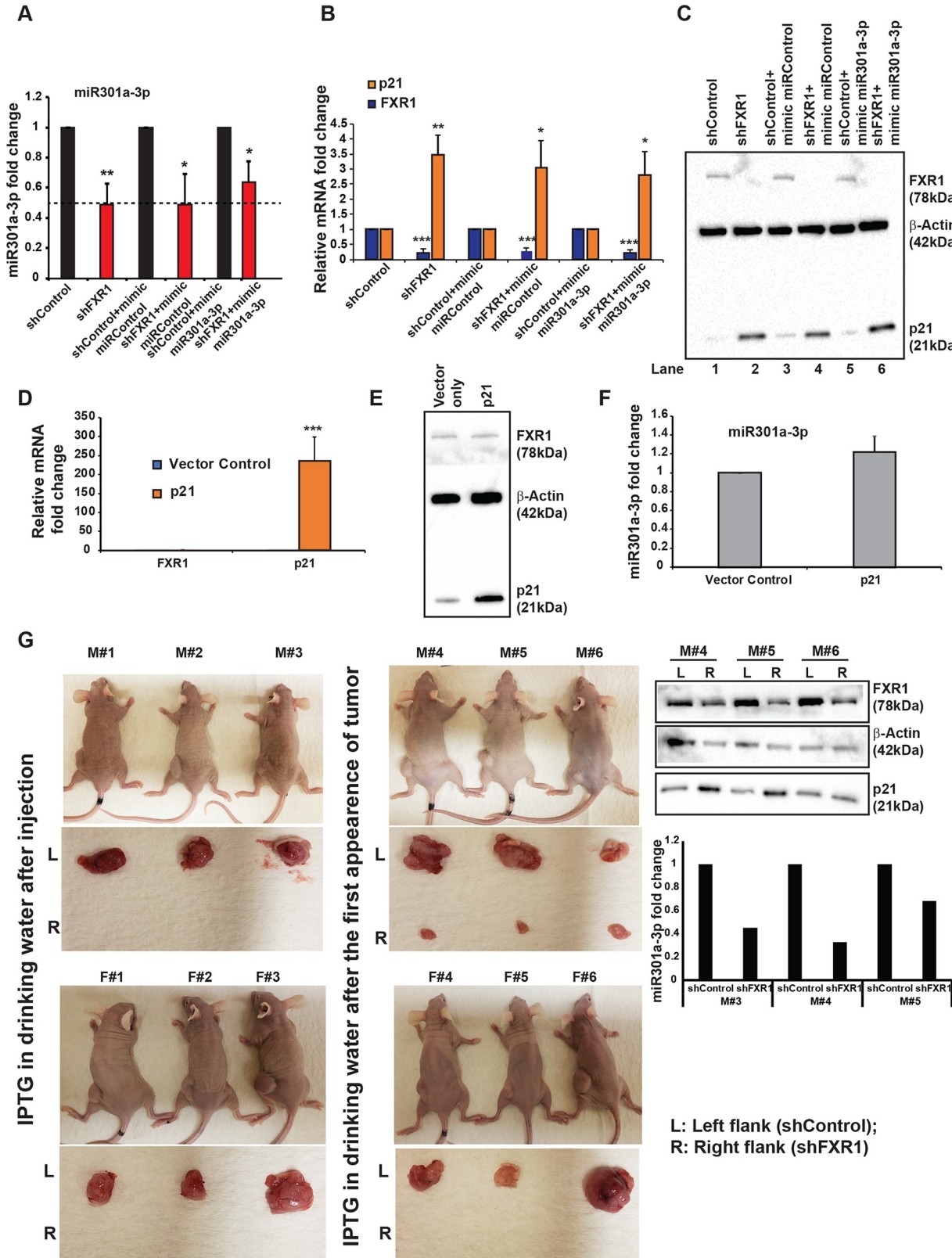

**Fig 6. FXR1 and miR301a-3p cooperatively target and repress p21 expression.** (**A**) qRT-PCR analyses of miR301a-3p in UMSCC74B cells treated with miRNA mimic and scrambled control in the presence and absence of FXR1. RNU6 serves as an endogenous control. (**B**) qRT-PCR

analyses of FXR1 and p21 in UMSCC74B cells treated with miRNA mimic and a scrambled control in the presence and absence of FXR1. *ACTIN* and *RPS18* serve as an endogenous control. (**C**) Western blot analyses of FXR1 and p21 from UMSCC74B treated with miRNA mimic and a scrambled control in the presence and absence of FXR1. β-Actin serves as a loading control. (**D**) qRT-PCR analyses of FXR1 and p21 in UMSCC74B cells transfected with p21 overexpression plasmid and vector control. *ACTIN* and *RPS18* serve as an endogenous control. (**E**) Western blot analyses of FXR1 and p21 from UMSCC74B cells transfected with p21 overexpression plasmid and vector control. β-Actin serves as a loading control. (**F**) qRT-PCR analyses of miR301a-3p in UMSCC74B cells transfected with p21 overexpression plasmid and vector control. RNU6 serves as an endogenous control. (**G**) Tumor image from 12 mice injected with UMSCC74B cells expressing IPTG inducible control and FXR1 shRNA clones and treated with 10mM IPTG/5% glucose in the drinking water. Tumors obtained from Male# 4–6 were used for western blot analyses for FXR1 and p21 where β-Actin serves as a loading control and qRT-PCR for miR301a-3p, RNU6 serves as an endogenous control. Data here represent the mean of n = 3 experiments. Statistical significance (*p*-value): * <0.05; ** <0.005; *** <0.0005.

suggest that FXR1 regulates the stability of miR301a-3p in oral cancer cells. First, FXR1 knockdown leads to a change in levels of many miRNAs both positively and negatively in oral cancer cells (Fig 1A and 1B), including decreased levels of miR29b-3p and miR301a-3p in multiple oral cancer cell lines (Fig 1C). Second, FXR1 knockdown leads to a decrease in miR301a-3p level when checked at different time points (Fig 3A). Third, FXR1 preferentially binds miR301a-3p and forms a miRNP complex to protect it from degradation (Figs 2A and 4E), which suggests that FXR1 plays a role in stabilizing miRNAs. These observations also indicate that FXR1 may act on some other miRNA substrates and promote their stability in oral cancer cells.

We provide evidence for the role of FXR1 in protecting miR301a-3p from degradation. RBPs form miRNP complexes and protect miRNAs from degradation. For example, RBPs ILF3 and BUD13 interact and stabilize miR-144 [13], as does RBP QKI, by forming a complex with miR-20 [47], indicating that the miRNA stability is linked to RBP-miRNA interactions. Herein, we suggest that FXR1 binds and protects miR301a-3p from exoribonuclease PNPT1-mediated degradation (Fig 4E). Furthermore, FXR1 is overexpressed in HNSCC and NSLSC [34, 35], along with miR301a-3p in multiple cancers [48–50]. The regulation of miR301a-3p stability by FXR1 may represent a novel pathway for stabilization of miR301a-3p in HNSCC.

## FXR1-dependent stabilization and PNPT1-mediated degradation modulate mature miR301a-3p levels in cancer cells

Our data suggest that FXR1 represses PNPT1-mediated degradation of miR301a-3p in oral cancer cells. FXR1-mediated stabilization of miR301a protects the miRNA from PNPT1 exonuclease activity. The level of miR301a-3p decreases upon FXR1 knockdown that can be rescued by a co-knockdown of PNPT1 (Fig 4B), suggesting that FXR1 blocks the activity of PNPT1 to promote the stability of miR301a. This mode of action of FXR1 enhances the levels of miR301a and its oncogenic functions in oral cancer cells. Similarly, another FXR family protein, FMRP, associates with miR-125b and miR-132 to deregulate NR2A and p250GAP and affect dendritic spine morphology [27]. These findings expand the role of FXR family proteins in the control of miRNAs in mammalian cells. Both miRNAs and RBPs can target and regulate specific mRNA transcripts through deadenylation, decapping, and degradation [14]. The mRNA deadenylase complex CCR4-NOT, decapping enzymes DCP1/2, and exonucleases XRN1/2 and PNPT1 have been implicated in both specific mRNA sequence-binding protein-mediated decay and in miRNA-mediated decay [51–54].

In mammals, the degradation of miRNAs is carried out by miRNases like XRN1, XRN2, RRP41, and PNPT1 [39, 42, 55]. We reveal that FXR1-mediated stability of miR301a-3p involves 3'-5' exonuclease PNPT1. PNPT1 3'-5' exonuclease activity can be blocked by RNA secondary structure or presence of RBPs at the 3'-end [56]. Similarly, we show that FXR1

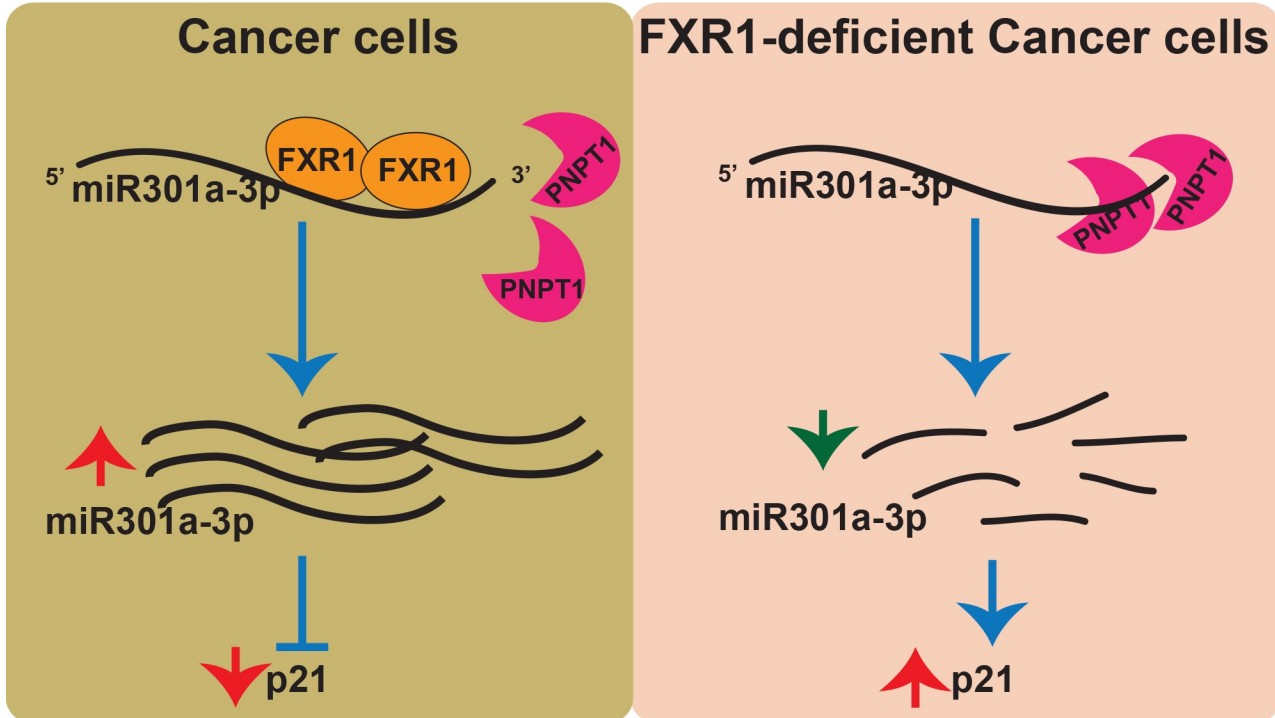

**Fig 7. FXR1 and miR301a-3p cooperatively target and repress p21 expression.** Model representation of FXR1 mediated stabilization of miR301a-3p from exonuclease PNPT1 to regulate p21 translation in cancer cells. Left: In cancer cells, FXR1 binds and protects mature miR301a-3p from exonuclease mediated degradation. FXR1-miRNA complex binds and degrades *p21* 3'-UTR. Right: In FXR1 depleted cancer cells, mature miR301a-3p is produced and degraded by 3'-5' exonuclease PNPT1. As a result, *p21* is stabilized in the cells resulting in a p21 protein upregulation.

blocks the exonuclease activity of PNPT1 to miR301a-3p by binding to the miRNA (Figs 2 and 4). Furthermore, knockdown of both FXR1 and PNPT1 rescues miR301a-3p (Fig 4).

## FXR1 overexpression causes downregulation of p21 signaling via a miR301a-3p pathway

Our previous findings suggest that mutating the FXR1 binding sites in the *p21* 3'-UTR fused to luciferase reporter, resulted in increased luciferase activity [34]. In this report, we find that mutating the miR301a-3p binding sites, that are also in the FXR1 binding region, in the *p21* 3'-UTR resulted in increased luciferase activity in the presence of FXR1, suggesting that both FXR1 and miR301a-3p are required to repress p21 in oral cancer cells.

Our analysis of the HNSCC TCGA database revealed that both miR301a-3p and FXR1 are significantly over-expressed in tumor samples (S1 and S3 Figs), signifying that the association between FXR1 and miR301a-3p may play a role in HNSCC cancer progression. MiR301a-3p is positively regulated by FXR1, stating that the levels of FXR1 protein and miR301a-3p are critical for repressing the tumor suppressor *p21* and promoting human oral and lung cancer. In the absence of FXR1, miR301a-3p is degraded by PNPT1, increasing p21 protein translation (Figs 1, 4 and 5), and in turn, promoting cellular senescence of the oral cancer cells [34]. Interestingly, miR301a-3p acts as an oncogene by targetting several tumor suppressor genes, including Smad4 in Laryngeal squamous cell carcinoma (LSCC) [48]. It will be beneficial to streamline further how the FXR1-miR301a-3p axis targets other genes to regulate their RNA stability, which may affect cancer progression.

The effect of FXR1 and miR301a-3p on p21 regulation suggests that genetic changes in the expression of these genes can induce oral cancer progression by decreasing the expression of p21. FXR1 overexpression that helps increase miR301a-3p-mediated repression of p21 possibly occurs in HNSCC to bypass senescence [34]. As expected, FXR1 overexpression is associated with poor prognosis of human oral cancers, according to TCGA analysis reported by us [34]. Conversely, our findings suggest that PNPT1 mediated degradation of miR301a in HNSCC can be protected by FXR1, leading to the repression of p21 aiding to poor prognosis of oral cancer. The effect of FXR1, PNPT1, and miR301a-3p expression suggests that the miRNA-mediated p21 regulation can be targeted to reduce the growth and proliferation of oral cancers.

## Materials and methods

### Cell lines, reagents, and antibodies

HOK cells (Sciencell#2610) were grown in keratinocyte serum-free medium supplemented with BPE and EGF (Gibco, BRL). HNSCC cell lines UMSCC11A, -74A and -74B were obtained from the University of Michigan and SCC4 (#CRL-1624), SCC25 (#CRL-1628), and Cal27 (#CRL-2095) were obtained from ATCC. Lung cancer cell line A549 was also obtained from ATCC. Cell lines UMSCC74B and Cal27 were routinely grown in Dulbecco's modified Eagle medium (DMEM) containing 10% fetal bovine serum (FBS) with 100U/ml penicillin-streptomycin (P/S). UMSCC11A and -74A were grown in DMEM containing 10% FBS, 100U/ml P/S, and 1X non-essential amino acids. SCC4 and SCC25 cell lines were grown in DMEM: F12 (1:1) containing 400 ng/ml hydrocortisone, 10% FBS, and 100U/ml P/S. A549 was grown in F-12K medium containing 10% FBS and 100U/ml P/S. Different shRNA constructs for FXR1 (TRCN0000158932 and TRCN0000159153) were obtained from Sigma Mission. miRI-DIAN miRNA hairpin human miR301a-3p inhibitor (IH-300657-05-0005), negative control (IN-001005-01-05), and miRIDIAN miRNA human miR301a-3p mimic (C-300657-03-0002) and negative control (C-300657-03-0002) were obtained from Dharmacon-GE Healthcare. DsiRNAs for PNPT1 (hs.RI.PNPT1.13.1, 13.2, and 13.3) were obtained from Integrated DNA Technologies (IDT). Human PNPT1 recombinant protein (ENZ-888) was purchased from ProSpec-Tany TechnoGene Ltd. P21 overexpression plasmid pCEP-WAF1 was a gift from Bert Vogelstein (Addgene plasmid # 16450) [57]. miRNA transcripts were purchased from Sigma-Aldrich. Antibodies: FXR1, Cell Signaling Technology (#12295, used predominantly for western blot), and EMD Millipore (#05–1529, used for IP and RNA-IP). From BD Pharmingen, p21 (#556431). From Proteintech, PNPT1 (14487-1-AP), GAPDH (10494-1-AP), Histone H3 (17168-1-AP), His-tag (66005-1-Ig) and β-Actin (60008-1-Ig). From Abcam, AGO2 (#ab57113). Horseradish peroxidase-conjugated anti-mouse and anti-rabbit immunoglobulin Gs were procured from GE Healthcare Biosciences (Uppsala, Sweden). Normal mouse (sc-2025) and rabbit (sc-2027) IgGs were obtained from Santa Cruz Biotechnology. Protein A/G plus (sc-2003) beads were purchased from Santa Cruz Biotechnology. LightSwitch Luciferase Assay kit (LS010) was purchased from Switchgear Genomics.

### miRNA-seq and analyses

shControl and shFXR1 treated UMSCC74B cells were used for total small RNA isolation followed by sequencing. The miRNA analysis was performed on the OnRamp Bioinformatics Genomics Research Platform (GRP), a high-throughput genomic analysis platform based on a Hadoop distributed storage system [58]. FASTQC [59] performed read quality control with the removal of regions with PHRED Score below 30. Cutadapt [60] scanned for the full set of TrueSeq adapters and removed them. The QC-screened and adapter removed reads were

input into a custom implementation of the CAP miRSeq pipeline [61]. In parallel, the reads were aligned via Bowtie [62] to hg19 genome to produce bam files, which were counted with htseq-counts [63] for RNA analysis and evaluated via GATK [64] to catalog miRNA coding variants. The reads passing QC were inputted to miRDeep2 [65] to produce a summary report and a table of known and novel miRNAs. Differential expression of miRNAs was evaluated by the edgeR [66] program with a threshold $Q_{adj}$ of 0.4.

## siRNA transfection and shRNA transduction

Cells were transfected with either 20nM DsiRNA and scrambled DsiRNA (IDT) or miRNA mimics and inhibitors with respective controls using Lipofectamine-2000 (Life Technologies# 11668019) following the manufacturer's protocol. Specific shRNA and control shRNA plasmids were used for the preparation of individual lentiviral particles. Cells were transduced with the lentiviral particles at an MOI (multiplicity of infection) of 25–50 in medium supplemented with 8μg/ml polybrene [34] and incubated for 72 hrs.

## Luciferase assays

The miR301a-3p antimiR and mimic, along with individual control-treated cells were used for luciferase assay. p21 full-length 3'-UTR and 3'-UTR with miRNA binding site mutations were cloned into a luciferase reporter vector system, pLightswitch-3'UTR from Switchgear Genomics. The segments were cloned between *Nhe*1 and *Xho*1 sites to express chimeric mRNAs. Luciferase GAPDH 3′-UTR and 3'-UTR empty vector negative controls were included in all assays. Each construct was transfected in UMSCC74B in triplicates separately with either miR301a-3p inhibitor/control inhibitor or miR301a-3p mimic/control mimic. Plates were incubated at 37˚C for 48 h post-transfection before being removed. p-LightSwitch Luciferase Assay was added to each experimental well of the 96 well solid bottom white plates and FXR1 represses PNPT1-mediated degradation of miR301a-3p at room temperature for 30 min. Luminescence was measured by using a VICTOR$^3$1420 Multilabel Counter (PerkinElmer).

## FXR1 RNA-IP (RIP)

FXR1 RNP IP is performed as previously described [67] with a few modifications. Briefly, an equal amount of total protein lysate is used (≥1mg) for each RIP. FXR1 monoclonal antibody (Millipore) or isotype control mouse IgG is pre-coated onto protein A/G plus agarose beads. Beads are extensively washed using NT2 buffer (50mM Tris–HCl, 150mM NaCl, 1mM MgCl$_2$, 0.05% Nonidet P-40 (NP-40), pH 7.4) supplemented with RNase inhibitors. Individual pull-down assays are performed at 4˚C for 2–4 h to minimize potential reabsorbing of mRNAs. For RNA analysis, the beads are incubated with 1mL NET2 buffer (1mL: 850μL NT2 with 10μL 0.1M DTT, 30μL 0.5M EDTA, RNase inhibitors) containing 20 U RNase-free DNase I (15 min, 30˚C), washed 3X with NT2 buffer and further incubated in 100μL NET2 buffer, 100μL proteinase K buffer (2X: 50mM Tris–HCl, 100mM NaCl, 20mM EDTA, 1% SDS, pH 7.4), and 0.5mg/mL proteinase K (30 min, 55˚C) to digest the proteins bound to the beads. RNA is extracted using Tri-Reagent.

## RNA extraction and qRT-PCR

Total RNA was prepared from the cell lines using Tri-Reagent by following the manufacturer's protocol. Small RNA (<200 bases) was isolated from the *mir*Vana miRNA isolation kit (Thermo Fisher #AM160) by following the manufacturer's protocol. 1 μg of total RNA was used for cDNA (mRNA) synthesis with the iScript kit (Biorad#1708891), and qRT-PCR of the

mRNA targets was performed using an Applied Biosystems StepOne Plus system with the 2X SYBR green master mix (Thermo Fisher# A25778). TaqMan miRNA RT kit (Thermo Fisher# 4366596) was used for miRNA cDNA preparation, followed by the Taqman Fast Universal 2X master mix (Thermo Fisher# 4444556). Primer sequences for mRNA genes are provided in the S1 Table. Custom TaqMan miRNA assays information is given in the S2 Table.

### Western blot analysis

Cells were lysed using RIPA buffer, supplemented with 1X protease inhibitor cocktail and PMSF, and proteins were separated using SDS-PAGE. Proteins were transferred to PVDF membrane, blocked in 5% skimmed milk, and incubated with primary antibodies at 4° C overnight. Membranes were washed three times with 1X Tris-buffered saline-0.1% Tween-20 and incubated with secondary antibody for 1 hour at room temperature. Proteins were visualized using substrates Clarity or Clarity Max (Biorad# 1705060 and 1705062), followed by Biorad Image Lab. For primary antibodies, the following dilutions were used: p21 and AGO2 (1:500), FXR1, Histone H3, and Dicer (1:1,000), PNPT1 (1:5,000), and GAPDH and β-Actin were used at 1:10,000 dilution.

### Northern hybridization

50 μg of total RNA was resolved on 12% urea-polyacrylamide gel and electronically transferred onto Zeta-probe GT membrane (Bio-Rad). Oligonucleotides complementary to miR301a-3p and snRU6 were 5'-end-labeled with [y-32P] ATP using T4 polynucleotide kinase (T4 PNK, New England Biolabs# M0201S) and used as probes for hybridization overnight at 42°C in Invitrogen Ambion ULTRAhyb Ultrasensitive Hybridization Buffer (AM8670). For the miR301a-3p probe, membranes were washed with 2X SSC with 0.1% SDS for 5 min, followed by 0.1X SSC with 0.1% SDS for 10 min to retain the signal. A phosphorimager screen was used to expose the radioactivity in the membrane, and Typhoon FLA9000 (GE Healthcare) imaging reader was used to measure the signal on the plate. snRU6, probed and 5S rRNA from the gel serve as a loading control [68]. The miR301a-3p signal was observed after 48–72 hr. of exposure.

### Cloning and purification of human FXR1 protein

Human FXR1 was PCR amplified and cloned in pET28a vector between the EcoR1/HindII sites. His-tagged protein was purified by $Ni^{2+}$- affinity chromatography and eluted using Imidazole using a method described elsewhere [69]. After purification, the fractions containing FXR1 (checked by SDS-PAGE followed by Coomassie staining) were pooled and dialyzed four times against a buffer containing 10 mM HEPES, pH 7.5, 300 mM LiCl, 5mM BME, 1mM EDTA, and 5% glycerol and variable imidazole concentrations, as follows: 200mM, 100mM, 50mM, 0mM at 4°C. Thrombin was used in the final dialysis buffer to cleave His-tag.

### Electrophoretic Mobility Shift Assay (EMSA)

Recombinant FXR1 or PNPT1 protein(s) were prepared/purchased and assembled onto miR301a-3p. 0.5 pmole of [y-32P] ATP labeled miRNA was mock-treated or mixed with recombinant FXR1 or PNPT1 or both (where mentioned) protein(s) and incubated at 37°C for 30 min. Reactions were carried out in the final volume of 20 μl containing 20 mM Na-HEPES, pH 7.0, 150 mM NaCl, 0.75 mM DTT, 1.5 mM EDTA and 10% glycerol with 1μg *E. coli* tRNA. After incubation, the samples were loaded onto 12% nondenaturing polyacrylamide gel containing 0.5X TBE (Tris-Cl, pH 8.0, Boric acid, EDTA). The electrophoresis was

performed at 4°C in 0.5X TBE for 6–8 h at 125 V. The RNA distribution or shift was visualized by autoradiography after gel drying.

## Inducible shRNA FXR1 xenograft mouse studies

Six-week-old male/female nude mice (Charles River Laboratories) were injected subcutaneously in the left flank with 5X10$^6$ UMSCC-74B cells stably transduced with control 3XlacO inducible shRNA and right flank of the same mouse with FXR1 3XlacO inducible shRNA (Sigma Aldrich) [34]resuspended in 20% Matrigel. One-day post-injection, 50% of the mice were treated with 10mM IPTG (Lab Scientific) and 5% glucose in the drinking water. The rest of the 50% of the mice were treated with 10mM IPTG/5% glucose in the drinking water after the tumor showed up. Tumor volumes were measured three times weekly using digital calipers and mice were sacrificed when tumor volumes exceeded 2000mm$^3$. Tumor volume formula = $0.5*(L*W^2)$ W = shortest side L = longest side.

## Statistical analysis

Data are expressed as the mean ± the standard deviation. Two-sample t-tests with equal variances are used to assess differences between means. Results with $p$ values less than 0.05 are considered significant.

## Supporting information

**S1 Fig. FXR1 alters the expression of a subset of miRNAs in oral and lung cancer cells.** (**A**) Box plot with one-way ANOVA of differential miRNA expression in the 604 patients obtained from the TCGA HNSCC database. Blue: primary tumor (n-483), green: metastatic tumor (n = 2), and brown: solid tissue normal (n = 44). X-axis signifies the sample type where y-axis plots the log-RPM (reads per million)-values of the differentially expressed miRNAs in HNSCC. (**B**) qRT-PCR of altered miRNAs in FXR1 KD UMSCC11A cells. RNU6 served as an endogenous control. (**C**) qRT-PCR of altered miRNAs in FXR1 KD UMSCC74A cells. RNU6 served as an endogenous control. (**D**) qRT-PCR of altered miRNAs in FXR1 KD A549 cells. RNU6 served as an endogenous control. (**E**) qRT-PCR is showing the KD efficiency *FXR1* in shFXR1 (used in Fig 1) treated cells used for miRNA analyses. Actin and GAPDH served as endogenous controls. (**F**) Western blot showing the KD efficiency of FXR1 by shRNA (TRCN0000158932) compared to a scrambled shRNA where β-Actin serves as a loading control. (**G**) qRT-PCR of altered miRNAs in FXR1 KD (TRCN0000158932) UMSCC74B cells. RNU6 served as an endogenous control. Data from B-D and F-G represent the mean of n = 3 experiments. Statistical significance (*p*-value): $^*<0.05$; $^{**}<0.005$; $^{***}<0.0005$.
(PDF)

**S2 Fig. Recombinant FXR1 protein purification and in vivo senescence assay. (A)** RNA-immunoprecipitation shows miRNA204-5p does not bind to FXR1 in UMSCC74B cells compared to control mouse IgG. Both *ACTIN* and *RPS18* served as endogenous controls. (**B**) Western blot analyses showing recombinant FXR1 protein expression before and after dialysis with the anti-His-tag antibody. (**C**) Western blot analyses showing recombinant FXR1 protein expression before and after dialysis with the anti-FXR1 antibody. (**D**) Beta-galactosidase assay showing, like the previous observation [34], instead of the miRNA alone, both miR301a-3p and TERC downregulation can induce senescence in UMSCC74B cells.
(PDF)

**S3 Fig. The stability of miR301a-3p is FXR1 dependent.** (**A**) qRT-PCR assay of FXR1 KD UMSCC11A cells showing significant down- and up-regulation of *FXR1* and *p21*, respectively,

compared to control. *AGO2* did not show any change after FXR1 KD. Both *ACTIN* and *RPS18* served as endogenous controls. (**B**) Western blot analyses of FXR1, p21, and AGO2 from UMSCC11A cells collected at different time points after FXR1 KD. GAPDH serves as a loading control. (**C**) qRT-PCR assay of FXR1 KD UMSCC11A cells showing significant miR301a-3p decay from 48 hrs compared to control. Cells were collected at the designated time points after shRNA transduction. RNU6 served as an endogenous control. (**D**) qRT-PCR assay of FXR1 and AGO2 KD UMSCC74B cells. Unlike FXR1 KD cells, *FXR1* and *p21* did not show any biologically relevant changes after AGO2 KD. Both *ACTIN* and *RPS18* served as endogenous controls. (**E**) Western blot analyses of FXR1, p21, and AGO2 from UMSCC74B cells after AGO2 KD. GAPDH serves as a loading control. (**F**) qRT-PCR assay of AGO2 KD UMSCC74B cells showing no significant regulation of miR301a-3p compared to control at 72hrs of transduction. RNU6 served as an endogenous control. Data here represents the mean of n = 3 experiments. Statistical significance (*p*-value): *<0.05; **<0.005; ***<0.0005.
(PDF)

**S4 Fig.** (**A**) Box plot with one-way ANOVA of differential miRNA expression in the 604 patients obtained from the TCGA HNSCC database. Blue: primary tumor (n = 520), green: metastatic tumor (n = 2), and brown: solid tissue normal (n = 44). X-axis signifies the sample type where y-axis plots the log-RPM (reads per million)-values of the differentially expressed miRNAs in HNSCC. (**B**) Western blot to test the KD of PNPT1 in UMSCC74B and UMSCC 11A cells with three siRNAs at 50nM. GAPDH serves as a loading control. (**C**) qRT-PCR analyses to test the expression of miR301a-3p in UMSCC11A cells under individual and double KD of FXR1 and PNPT1. RNU6 served as an endogenous control. (**D**) qRT-PCR analysis of *FXR1*, *p21*, and *PNPT1* in UMSCC11A cells under individual and double KD of FXR1 and PNPT1. Both *ACTIN* and *RPS18* served as endogenous controls. (**E**) Western blot analysis of FXR1, p21, and PNPT1 in UMSCC11A cells under individual and double KD of FXR1 and PNPT1. GAPDH serves as an endogenous control. (**F**) EMSA shows that both rFXR1 and rPNPT1 proteins are unable to bind and degrade, respectively, the in vitro transcribed miR204-5p. Data here represent the mean of n = 3 experiments. Statistical significance (*p*-value): *<0.05; **<0.005; ***<0.0005.
(PDF)

**S5 Fig. miR301a-3p targets *p21* mRNA and reduce its expression.** (**A**) qRT-PCR analyses to test the expression of miR301a-3p in UMSCC74A cells treated with miRNA inhibitor with scrambled control. RNU6 served as an endogenous control. (**B**) p21 protein is up-regulated in miR301a-3p inhibitor transfected UMSCC74A cells. β-Actin serves as a loading control. (**C**) *p21* 3'UTR luciferase activity is significantly up-regulated in the presence of miR301a-3p inhibitor in UMSCC74A cells compared to the scrambled control transfected cells. Forty-eight hours after transfection of UMSCC74A cells with miRNA control and 301a-3p inhibitor along with empty 3'-UTR luciferase plasmid and wild type *p21* 3′-UTR, the lysates were analyzed for luciferase activity using a luminometer. The empty 3'UTR luciferase plasmid served as a transfection and loading control. Values are the means ± SD from three independent experiments by using an unpaired two-sample t-test. (**D**) Expression of miR301a-3p in UMSCC74A cells treated with miRNA mimics. RNU6 served as an endogenous control. (**E**) p21 protein is down-regulated in miR301a-3p mimic treated UMSCC74A cells. β-Actin serves as a loading control. (**F**) *p21* 3'-UTR (full-length wild type and mutated miRNA binding sites) luciferase activity with an expression of miR301a-3p mimic in UMSCC74A cells. In the presence of miR301a-3p mimic, the p21 3'-UTR luciferase activity significantly reduces whereas the mutants show a highly significant up-regulation. Experiments were performed as described in (**C**). (**G**) qRT-PCR analyses to test the expression of miR301a-3p in A549 cells treated with

miRNA inhibitor with scrambled control. RNU6 served as an endogenous control. (**H**) p21 protein is up-regulated in miR301a-3p inhibitor transfected A549 cells. GAPDH serves as a loading control. (**I**) *p21* 3'UTR luciferase activity is significantly up-regulated in the presence of miR301a-3p inhibitor in A549 cells compared to the scrambled control transfected cells. Experiments were performed as described in (C). The empty 3'UTR luciferase plasmid served as a transfection and loading control. Values are the means ± SD from three independent experiments by using an unpaired two-sample t-test. (**J**) Expression of miR301a-3p in A549 cells treated with miRNA mimics. RNU6 served as an endogenous control. (**K**) p21 protein is down-regulated in miR301a-3p mimic treated A549 cells. GAPDH serves as a loading control. Data here represents the mean of n = 3 experiments. Statistical significance (*p*-value): *$<0.05$; **$<0.005$; ***$<0.0005$.
(PDF)

**S1 Table. Primers used in the study are listed in the S1 Table.**
(DOCX)

**S2 Table. miRNA primers/probes used in the study are listed in the S2 Table.**
(DOCX)

## Author Contributions

**Conceptualization:** Mrinmoyee Majumder, Viswanathan Palanisamy.

**Data curation:** Mrinmoyee Majumder, Viswanathan Palanisamy.

**Formal analysis:** Mrinmoyee Majumder, Viswanathan Palanisamy.

**Funding acquisition:** Viswanathan Palanisamy.

**Investigation:** Mrinmoyee Majumder, Viswanathan Palanisamy.

**Methodology:** Mrinmoyee Majumder, Viswanathan Palanisamy.

**Project administration:** Viswanathan Palanisamy.

**Resources:** Viswanathan Palanisamy.

**Supervision:** Viswanathan Palanisamy.

**Validation:** Viswanathan Palanisamy.

**Writing – original draft:** Mrinmoyee Majumder, Viswanathan Palanisamy.

**Writing – review & editing:** Mrinmoyee Majumder, Viswanathan Palanisamy.

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
