## [Decision Letter · Decision Letter 0]

9 Oct 2019

Dear Dr Palanisamy,

Thank you very much for submitting your Research Article entitled 'RNA Binding Protein FXR1-miR301a-3p axis contributes to p21WAF1 degradation in oral cancer' to PLOS Genetics. Your manuscript was fully evaluated at the editorial level and by independent peer reviewers. The reviewers appreciated the attention to an important problem, but raised some substantial concerns about the current manuscript. Based on the reviews, we will not be able to accept this version of the manuscript, but we would be willing to review again a much-revised version. We cannot, of course, promise publication at that time.

While all authors recognize the novelty, innovation, and importance of the subject, there were concerns with the inclusion of appropriate controls for the EMSA and other experiments.  It needs to be determined if FXR1 directly alters PNPT1 to determine effects on miRNA degradation, performance of experiments in the presence of transcription inhibitors, and a direct binding of PNPT1 to miR301.  It is also strongly recommended to clarify the binding site proximity for FXR1 and miR301 in the p21 3'UTR.  Finally, Reviewer #2 brings up a good point in that cellular experiments should be performed to validate the proposed molecular axis.  This would add physiological validity to the model.

If you decide to revise the manuscript for further consideration at PLOS Genetics, please aim to resubmit within the next 60 days, unless it will take extra time to address the concerns of the reviewers, in which case we would appreciate an expected resubmission date by email to plosgenetics@plos.org.

[LINK]

We are sorry that we cannot be more positive about your manuscript at this stage. Please do not hesitate to contact us if you have any concerns or questions.

Yours sincerely,

Michael V. Autieri

Guest Editor

PLOS Genetics

David Kwiatkowski

Section Editor: Cancer Genetics

PLOS Genetics

Reviewer's Responses to Questions

**Comments to the Authors:**

Reviewer #1: In this manuscript, the authors reported that a RNA-binding protein FXR1 control miR301a-3p stability by directly binding to this miRNA and prevent its degradation by PNPT1 in HNSCC. As miR301a targets p21, thereby FXR1 promote the growth and proliferation of HNSCC. The report does contain some innovative findings. The manuscript is well written and all of data are beautifully presented. However, all of data were collected on the cultured cell lines. No data show there is functional relevance in vivo. The reviewer think the report is still preliminary and need to add some in vivo observation.

Reviewer #2: The manuscript “RNA binding protein FXR1-miR301a-3p axis contributes to p21WAF1 degradation in oral cancer” provides a compelling case that FXR1 protects miR301a-3p from degradation from exonuclease PNPT1 which allows the miR to destabilize p21, forming a novel axis of regulation in oral cancer cells. Overall, I find this manuscript to be of great interest to the fields of both RBP-miR interactions and RNA regulation of cancer biology.

Major Critiques:

1. Figure 2A needs a negative control gene in addition to the IgG control shown for FXR1-miR 301a-3p interaction by pulldown.

2. Figure 2 shows an EMSA gel shift experiment, but the figure would be more convincing and enhanced using a negative control probe to demonstrate specificity in FXR1 binding in vitro.

3. Figure 4E again would be enhanced by a negative control probe especially due to degradation of miR301a-3p probe by PNPT1 to again show specificity of the interaction and degradation.

4. Although it is clear FXR1 binds miR301a-3p, can you rule out that FXR1 directly alters PNPT1 to directly or indirectly prevent miR degradation?

5. The overall interaction axis between FXR1-miR301a-3p-p21 suggests effects on cellular senescence and proliferation, and experiments measuring changes in the cell would strengthen the paper demonstrating tangible effects of this interaction in cancer cells. These cellular experiments should be performed manipulating participating components of the novel axis to determine effects on cellular function.

Minor Critiques:

1. The paper introduces and shows data involving the FXR1-miR29b-3p interaction but fails to explain why this was not continued.

2. Discussion points: Does miR 301a-3p target any other cancer or senescence related transcripts other than p21 based on the target sequence? Further, does FXR1 bind p21 directly and regulate the transcript by RNA stability, and how that might affect the role of FXR1 in the axis identified.

Reviewer #3: Majumder and Palanisamy describe a novel and very relevant regulatory model involving miR-301a and the RNA binding protein FXR1. The data is convincing and supports their model. However, there are a few weak points and further analysis should be done to expand their conclusions and increase the value of their article.

Major points

-Changes in transcription levels could interfere with results. Transcription should be shut down, then they can properly evaluate the impact of FXR1 on miR-301a stability.

-In Figure 2C, it would be good to have a negative control with a mutated miR-310a.

-There is probably CLIP data available for FXR1. The authors should check if miR-310a is described as a binding target and determine the sequence that is recognized.

-Binding of PNPT1 to miR-301a is not shown.

-In fact, their model can have two components and they have to check that. It looks like that binding and function of FXR1 to p21 transcript is influenced by miR301 binding and vice versa. Are miR301a and FXR1 binding sites in p21 3'UTR close by? They can check for instance using luciferase constructs if mutation of miR301a binding site in p21 3'UTR abolishes FXR1 binding and regulation. Similarly, they can mutate FXR1 site and determine if miR301a regulation is affected.

Minor points:

-The authors should check the expression correlation between FXR1 and miR-301a in TCGA head and neck cancer and impact on survival.

-The introduction is somehow confusing and goes back and forth talking about miRNAs and RBPs.

-The authors should try to go beyond their model. This could really increase the value of their article. miRNAs and RBPs can be part of regulatory networks, for instance Musashi1 and miR-137 (31004009). Using available data and predictions, they could check if FXR1 and mir301a co-regulate other target genes.

**Have all data underlying the figures and results presented in the manuscript been provided?**

Reviewer #1: Yes

Reviewer #2: Yes

Reviewer #3: Yes

PLOS authors have the option to publish the peer review history of their article (what does this mean?). If published, this will include your full peer review and any attached files.

Reviewer #1: No

Reviewer #2: No

Reviewer #3: No

---

## [Decision Letter · Decision Letter 1]

20 Dec 2019

Dear Dr Palanisamy,

We are pleased to inform you that your manuscript entitled "RNA Binding Protein FXR1-miR301a-3p axis contributes to p21WAF1 degradation in oral cancer" has been editorially accepted for publication in PLOS Genetics. Congratulations!

Yours sincerely,

Michael V. Autieri

Guest Editor

PLOS Genetics

David Kwiatkowski

Section Editor: Cancer Genetics

PLOS Genetics

Comments from the reviewers (if applicable):

Reviewer's Responses to Questions

**Comments to the Authors:**

Reviewer #2: I am satisfied by the revisions both in the results and discussion sections.

Reviewer #3: Authors successfully addressed my questions and I think the paper is ready for publication.

**Have all data underlying the figures and results presented in the manuscript been provided?**

Reviewer #2: Yes

Reviewer #3: None

PLOS authors have the option to publish the peer review history of their article (what does this mean?). If published, this will include your full peer review and any attached files.

Reviewer #2: No

Reviewer #3: No

**Data Deposition**

http://datadryad.org/submit?journalID=pgenetics&manu=PGENETICS-D-19-01404R1

**Press Queries**

---

## [Editor Report · Acceptance letter]

8 Jan 2020

PGENETICS-D-19-01404R1 

RNA Binding Protein FXR1-miR301a-3p axis contributes to p21^WAF1^ degradation in oral cancer 

Dear Dr Palanisamy, 

We are pleased to inform you that your manuscript entitled "RNA Binding Protein FXR1-miR301a-3p axis contributes to p21^WAF1^ degradation in oral cancer" has been formally accepted for publication in PLOS Genetics! Your manuscript is now with our production department and you will be notified of the publication date in due course.

With kind regards,

Matt Lyles

PLOS Genetics

On behalf of:
